# Safety and Effectiveness of Perioperative Hyperthermic Intraperitoneal Chemotherapy with Gemcitabine in Patients with Resected Pancreatic Ductal Adenocarcinoma: Clinical Trial EudraCT 2016-004298-41

**DOI:** 10.3390/cancers16091718

**Published:** 2024-04-28

**Authors:** David Padilla-Valverde, Raquel Bodoque-Villar, Esther García-Santos, Susana Sanchez, Carmen Manzanares-Campillo, Marta Rodriguez, Lucia González, Alfonso Ambrós, Juana M. Cano, Maria Padilla-Marcote, Javier Redondo-Calvo, Jesus Martin, Leticia Serrano-Oviedo

**Affiliations:** 1Head of the Hepatobiliary Surgery Unit and Carcinomatosis Programme, Department of Surgery, General University Hospital, Faculty of Medicine, UCLM, C/Obispo Rafael Torija s/n, 13005 Ciudad Real, Spain; esther_garcia_santos@hotmail.com (E.G.-S.); susisg_83@hotmail.com (S.S.); carmenmc2010@gmail.com (C.M.-C.); marcote1528@gmail.com (M.P.-M.); jesusm@sescam.jccm.es (J.M.); 2Traslational Investigation Unit, University General Hospital of Ciudad Real, SESCAM, Research Institute of Castilla-La Mancha (IDISCAM), C/Obispo Rafael Torija s/n, 13005 Ciudad Real, Spain; raquel_t_n@hotmail.com (R.B.-V.); fjredondo@sescam.jccm.es (J.R.-C.); 3Department of Pharmacy, General University Hospital, Ciudad Real, Faculty of Medicine, UCLM, C/Obispo Rafael Torija s/n, 13005 Ciudad Real, Spain; martarm5@hotmail.com; 4Department of Pathology, General University Hospital, Ciudad Real, Faculty of Medicine, UCLM C/Obispo Rafael Torija s/n, 13005 Ciudad Real, Spain; luciagonlope@gmail.com; 5Intensive Care Unit, General University Hospital, Ciudad Real, Faculty of Medicine, UCLM, C/Obispo Rafael Torija s/n, 13005 Ciudad Real, Spain; alfosoa@sescam.jccm.es; 6Oncology Department, University General Hospital, Ciudad Real, Faculty of Medicine, UCLM, C/Obispo Rafael Torija s/n, 13005 Ciudad Real, Spain; juanamariacano@gmail.com

**Keywords:** pancreatic ductal adenocarcinoma, HIPEC, chemohyperthermia, gemcitabine, pancreatic cancer stem cells

## Abstract

**Simple Summary:**

Despite the massive therapeutic advances made and the huge interest in discovering the characteristics of pancreatic cancer and its biological behavior, it continues to present a devastating prognosis. One of the characteristics of this disease is its intense capacity for locoregional invasion, increasing the capacity for recurrence and, therefore, the mortality of the patient. Recently, a population of cells known as pancreatic cancer stem cells, capable of self-renewal and differentiation and highly resistant to conventional therapy, has been identified as the origin of pancreatic cancer. Due to the above, the use of HIPEC with gemcitabine after cytoreductive surgery in patients with pancreatic ductal adenocarcinoma could decrease locoregional recurrence and improve prognosis by eliminating residual abdominal pancreatic cancer stem cells.

**Abstract:**

Background: Despite the improvement in therapies, pancreatic cancer represents one of the most cancer-related deaths. In our hypothesis, we propose that Hyperthermic Intraperitoneal Chemotherapy with gemcitabine after pancreatic cytoreductive surgery could reduce tumor progression by reducing residual neoplastic volume and residual pancreatic cancer stem cells. Materials and methods: A randomized trial involving 42 patients. All patients were diagnosed with pancreatic ductal adenocarcinoma. Group I: R0 resection. Group II. R0 resection and HIPEC with gemcitabine (120 mg/m^2^ for 30 min). Effectiveness was measured with analysis of overall survival, disease-free survival, distant recurrence, locoregional recurrence, and measuring of pancreatic cancer stem cells (EpCAM^+^CXCR4^+^CD133^+^). Results: From 2017 to 2023, 63 patients were recruited for our clinical trial; 21 patients were included in each group, and 21 were excluded. Locoregional recurrence, *p*-value: 0.022, was lower in the experimental group. There were no significant differences between the two groups in hospital mortality, perioperative complications, or hospital costs. We found a significant decrease in pancreatic cancer stem cells in patients in the experimental group after treatment, *p* -value of 0.018. Conclusions: The use of HIPEC with gemcitabine after surgery in patients with resectable pancreatic ductal adenocarcinoma reduces locoregional recurrence and may be associated with a significant decrease in pancreatic cancer stem cells.

## 1. Introduction

In recent years, therapeutic advances in pancreatic ductal adenocarcinoma have not improved the poor prognosis, and it represents the fourth leading cause of cancer mortality in developed countries [1,2,3]. Surgery is the only curative treatment for only 15–20% of the patients that can be initially resected because of the high percentage of patients with an unresectable locoregional or distant disease at diagnosis [2,3,4,5,6]. We should consider that there may be other mechanisms that allow disease recurrence to appear even when extensive oncologic procedures are performed.

A population of pancreatic cancer stem cells, PaCSCs, has been identified with a high capacity for malignant transformation. They have a high rate of self-renewal, the ability to develop cell subtypes, and be highly resistant to conventional therapy [7,8,9,10,11,12,13,14,15,16,17,18,19]. Upon contact with these drugs, PaCSCs are prevented from proliferating and enter a dormant state. Later, following chemotherapy, these cells undergo a significant increase in proliferative activity, enabling the recurrence of the disease [12,13]. Their intraoperative identification is not performed, so we could be overstaging the surgery and understaging the disease. Today the identification of surgical margins and intraoperative cytology, with perioperative radiological methods, identify the radical nature of the surgery performed, but perhaps they may not be sufficient.

We have developed a new therapeutic model as an adjuvant treatment in patients with pancreatic cancer, distinguished by the application of Hyperthermic Intraperitoneal Chemotherapy, HIPEC, with gemcitabine to eliminate the early proliferation and the locoregional tumor invasion of pancreatic cancer stem cells and improve prognosis. The use of HIPEC has been studied in different experimental models with or without peritoneal carcinomatosis of various origins and with different chemotherapy drugs, achieving encouraging results in terms of locoregional recurrence and overall survival [20,21,22,23,24,25,26,27,28,29,30]. This study involves a prospective randomized clinical trial involving patients with pancreatic ductal adenocarcinoma operated on in the surgery department of the General University Hospital of Ciudad Real. We hypothesized that HIPEC, with the use of gemcitabine after cytoreductive surgery, will decrease tumor progression of pancreatic cancer by reducing the residual neoplastic volume and PaCSCs subpopulation (EpCAM^+^CXCR4^+^CD133^+^), improving patient survival by reducing disease recurrence. Our aim was to explore the effectiveness in terms of recurrence, overall survival, and disease-free survival, as well as to study the perioperative morbidity of the experimental group with cytoreductive surgery and HIPEC with gemcitabine compared to the conventional group without HIPEC. Finally, we characterized the presence of PaCSCs (EpCAM^+^CXCR4^+^CD133^+^) in the experimental group before and after treatment with surgery and HIPEC with gemcitabine.

## 2. Material and Methods

### 2.1. Population

This is a phase II–III randomized, single-blind clinical trial that included a population of forty-two patients, twenty-one patients in each group, accepting an alpha risk of 0.05 and a beta risk of 0.2 in a bilateral comparison. They were operated on in the surgery department of the General University Hospital of Ciudad Real, and tumors were resected with curative intent from 2017 to 2022. We had to extend the recruitment period to 2023 due to the SARS-CoV-2 pandemic. For the survival study, there is an extended follow-up of at least a median of 18 months. Patients were randomized into two groups by software, and a number was assigned to the patient by the principal investigator. After explaining the treatment and the single-blind clinical trial to the patients, we obtained their informed consent. All patients included in the study had an R0 resection in which the distance between the tumor and the surgical margin was greater than 1 mm. The study was approved by the clinical research ethics committee of our hospital, and the investigations were carried out following the rules of the Declaration of Helsinki (revised in 2013):

Group I: After an R0 resection, patients received an individualized adjuvant treatment.

Group II: After R0 resection, HIPEC was performed with gemcitabine (120 mg/m^2^ for 30 min), and an individualized adjuvant treatment was considered.

### 2.2. Exclusion Criteria

In reference to the exclusion criteria, we rejected those patients who did not wish to participate in the trial; patients with locoregional or distant disease that contraindicated surgical treatment, diagnosed preoperatively or intraoperatively; patients with neoadjuvant treatment and patients with the presence of a synchronous neoplastic disease. In addition, the absence of cardiorespiratory, renal, or hepatic insufficiency was required in these patients. Finally, women of childbearing age had to have a negative serum or urine pregnancy test result at the screening visit.

### 2.3. Data Collection

The following variables were collected in this study: Main Variables: Overall survival, disease-free survival, locoregional recurrence, distant recurrence, PaCSCs. Other variables: Clavien-Dindo system used for grading complications. Pancreatic surgical complications: pancreatic fistula, bleeding, delayed gastric emptying. Operative mortality (within thirty days after surgery). Clinical and surgical variables: age, sex, diabetes, dyslipidemia, arterial hypertension, symptomatology, surgical treatment, adjuvant treatment, chemotherapy regimen, operative time, and hospital stay. Histologic variables: tumor grade (G1, G2, G3), neurologic, vascular and lymphatic invasion, pathologic nodes. Biochemical and gasometric parameters to control system functions were measured three times: preoperatively, 24 hours after surgery, and on the seventh postoperative day. Intra-hospital costs of the patients.

### 2.4. Surgical Procedure

We performed a bilateral subcostal laparotomy, and after an examination of the abdominal cavity to identify liver or peritoneal metastases that would contraindicate the resection, we realized a retroperitoneal first access to the superior mesenteric artery. Depending on the location and perioperative morphologic characteristics of the primary pancreatic ductal adenocarcinoma, we carried out a cephalic pancreaticoduodenectomy, distal, non-anatomic, or total pancreatectomy.

### 2.5. HIPEC

During the surgery, after an intraoperative histological diagnosis of pancreatic ductal adenocarcinoma was made and the radical surgical resection was performed, we used a closed HIPEC system with CO_2_ recirculation (PRS Combat^®,^, Galmaz Biotech SL, Madrid, Spain). This model uses an external heater and a carrier solution pump. This system includes a gas exchanger that allows us to control the filling of the peritoneal cavity with the drug solution and the output of the CO_2_ that is used to create turbulence for complete drug distribution. During the recirculation of gemcitabine, 120 mg/m^2^ for 30 min, CO_2_ creates intra-abdominal turbulence (Figure 1).

### 2.6. Isolation of PaCSCs

We collected samples of intraperitoneal ascitic fluid before the start of surgical resection and a second sample after HIPEC in the last intra-abdominal circulating solution, called pre-HIPEC and post-HIPEC, respectively. The resulting pellet after centrifugation (1100 rpm) of the pre-HIPEC and post-HIPEC intraperitoneal ascitic fluid samples was resuspended in 1 mL of warm PBS and blocked for 15 min on ice with Flebogamma Dif 100 mg/mL (Grifols S.A., Barcelona, Spain). After that, samples were washed with PBS, centrifuged again, and the cellular pellet was resuspended in 100 µL PBS and incubated with the corresponding antibodies (1 µL /100 µL PBS) for 30 min on ice in darkness. Finally, the cells were washed with PBS, centrifuged, and the resulting pellet was resuspended in 500 µL of PBS. Flow cytometry was used to identify pancreatic cancer stem cell surface markers, EpCAM, CXCR4, and CD133 (Miltenyi Biotec MAQSquant® Analyzer cytometer, Bergisch Gladbach, Germany). The antibodies used were anti-PE-EpCAM (clone: REA764, Myltenyi Biotec, Auburn, CA, USA), anti-PE-Vio770-CXCR4 (clone: REA649, Miltenyi Biotec, USA), and anti-APC-CD133 (clone: REA753, Miltenyi Biotec, USA).

### 2.7. Statistical Analysis

Frequency distributions were obtained for qualitative and quantitative variables. Qualitative variables were expressed as counts and frequencies, *n* (%). Quantitative variables were expressed as the mean, median, and standard error of the mean (SEM) and compared using the Student *T* or Mann–Whitney test. Correlations between the different parameters were analyzed using the Pearson correlation method. Cumulative survival rates were calculated by the Kaplan–Meier method, and differences in survival rates were analyzed by the Log-rank test. *p*-values < 0.05 were considered significant differences. Statistical analysis was conducted using SPSS 29.0 (IBM SPSS, Armonk, NY, USA), and the graphical representations were displayed with GraphPad Prism 9.5.1 (GraphPad Software, Boston, MA, USA).

## 3. Results

From 2017 to 2023, 63 patients with a suspected diagnosis of pancreatic ductal adenocarcinoma were recruited for this clinical trial. There were 21 patients who were excluded because of intraoperative unresectability or different intraoperative histological diagnoses. The remaining 42 patients were randomly divided into Group I (*n* = 21) and Group II (*n* = 21) (Figure 2). The median age was 68 years (range 42–86). Nineteen patients were male (45.2%) and twenty-three were female (54.8%). All the characteristics of the patients are listed in Table 1.

Regarding group II, the majority of cases had no postoperative complications, 9 patients (42%), or had some minor postoperative complications treated medically without requiring surgery or an endoscopic approach [grade I complications, 4 (19%), and grade II complications, 5 (24%)]. Only 2 patients (10%) had major grade III complications. Group I patients had some minor postoperative complications [grade I, 3 patients (14%), and grade II, 3 (14%)]. Six patients (29%) had major grade III and IV complications. Intrahospital mortality was equal in both groups (5%), with only one patient per group. Ultimately, no statistically significant differences in postoperative complications were observed between the groups or the adjuvant treatment regimen (Table 1). The median length of hospital stay was 17 days (range 4–69) in Group I. In Group II, the median hospital stay was 11 days (range 7–28). There were no statistically significant differences in length of hospital stay between the groups (Table 1).

There was no significant difference between the values of PaCSCs, measured in the experimental group, 19.7 ± 8.2, with respect to the control group, 13.8 ± 7.1. Regarding the values before and after HIPEC (19.7 ± 8.2 and 5.8 ± 31.1, respectively) in the experimental group, a significant reduction was observed, *p*-value 0.018 (Table 1).

The preoperative biochemical and gasometric parameters, 24 h after intervention and 7 days after the intervention, are shown in supplementary info (Appendix A). The exceptional statistically significant differences between group I and group II did not affect the functionality of the systems studied, so the application of chemohyperthermia could be considered a safe technique. They can be explained as regulatory mechanisms after surgery with a tendency to normalize in the last control.

The mean cost of the patients during their stay in our department, using cost per stay and GRD (Diagnosis Related Group), for each patient, was 14,979.24 euros, in the experimental group, with respect to the control group, 21,220.5 euros, with no significant differences.

At a median follow-up of 18 months, there were no significant differences between the two groups with regard to distant recurrence (lung and liver), overall survival, or disease-free survival. However, there were differences regarding the locoregional recurrence of the disease, *Log-rank p* = 0.022 (Figure 3).

## 4. Discussion

The prognosis for pancreatic cancer is unfavorable. Despite advances in diagnosis and treatment, pancreatic cancer remains one of the deadliest malignancies. Upon diagnosis, patients often exhibit either locoregional recurrence (30–40%) or distant metastases (50%). Surgery is the only potentially curative option for these patients. Currently, surgical resection is the only potentially curative technique to treat this disease. However, more than 80% of patients have advanced-stage disease at diagnosis, limiting the possibilities for complete surgical resection of the tumor, with a five-year survival rate of no more than 10% [31].

PaCSCs play a major role in cancer invasion, recurrence, metastasis, and drug resistance because these cells have the ability to self-renew and can lead to tumorigenesis. Mechanisms of drug resistance include cellular plasticity and, in particular, the capacity of CSCs to adopt a quiescent state, increased capacity DNA repair capacity, epithelial-mesenchymal transition (EMT), EMT-type cells, tumor microenvironment, and the involvement of multiple signaling pathways [12,13,14,15,16,17,18,19].

These pancreatic cancer stem cells could be located in the peritoneum during surgery, as well as tumor cells could detach from the tumor and remain loose and be responsible for recurrence. Sugarbaker et al. observed that HIPEC is effective in eradicating microscopic deposits of residual cancer both at the resection site and on the peritoneal surface [32]. The analysis of the presence of cancer cells in the ascitic liquid is not an effective technique because there are many false negative tests or the tumor cells that have been identified do not have the capacity to proliferate because they are not cancer stem cells.

Peritoneal carcinomatosis can manifest as an early locoregional stage, persisting within the abdominal cavity before evolving into a distant spread of the disease. By enhancing the locoregional treatment approach for pancreatic cancer through radical surgery and perioperative adjuvant HIPEC, there is potential to eliminate residual disease. Moreover, the additional use of hyperthermia to intraperitoneal chemotherapy could enhance the locoregional therapeutic effect by provoking a direct “toxic shock” on pancreatic cancer stem cells and promoting a greater penetration of cytotoxic drugs into the tissues [33].

Historically, HIPEC has been studied in several experimental models of peritoneal metastasis in different types of cancer, with safe and effective results in disease reduction. Many of these peritoneal models have included gemcitabine to determine its pharmacokinetic and pharmacodynamic behavior, such as the study by Morgan et al. in which, in patients with peritoneal carcinomatosis of different causes, they identified the tolerated dose, toxicity, and pharmacokinetic characteristics of gemcitabine, and recommended its use at 120 mg/m^2^ [26]. Gamblin et al. administered doses of 200 mg/m^2^, which were well-tolerated in patients with unresectable pancreatic ductal adenocarcinoma [27]. Years later, Tentes et al. utilized gemcitabine in patients with pancreatic cancer and peritoneal metastases. Due to the limited number of participants and the varying gemcitabine doses, definitive conclusions could not be reached [30].

Later, in a phase II study of adjuvant intraperitoneal gemcitabine administration in patients with resectable pancreatic ductal adenocarcinoma, the procedure lasted a total of 60 min using the open abdominal technique, and after surgery using a Port-A-Cath. The authors also concluded that intraperitoneal administration of the drug was well tolerated, with no documented complications [34,35].

In 2012, Tentes et al. included patients with limited macroscopic peritoneal metastases and R1 cytoreductive surgery, but patients were not randomized. In this case, HIPEC was performed for 60 min at 42–43 °C with Gemcitabine at a dose of 1000 mg/m^2^, using the open Coliseum technique. Preliminary results showed a survival benefit in those patients undergoing CRS with HIPEC and no increase in morbidity and mortality from intraoperative Gemcitabine. They extended their study and, in 2018, confirmed that CRS with HIPEC could be considered as a treatment option in highly selected patients with pancreatic cancer and peritoneal carcinomatosis [36,37].

Because there was no homogeneity in the medical literature about the dose of gemcitabine, we should use and the identification of PaCSCs as the possible etiology of recurrence, we developed a prototype model aimed at discerning the toxicity and effectiveness of HIPEC using gemcitabine.

In this previous rat model of our group, we observed a significant reduction of pancreatic cancer stem cells, which we determined by CD133^+^CXCR4^+^ labeling [38].

As a result of this study, our group initiated the first randomized clinical trial to determine the approaches to the use of this therapy in pancreatic ductal cancer. The model we performed was a closed technique characterized by recirculation of the drug gemcitabine (120 mg/m^2^) in a hyperthermic solution called Physioneal 40 for 30 min with CO_2_ agitation and controlled temperature between 41–42 °C. In this way enhances the tissue penetration of chemotherapy and minimizes heat loss and the risk of contamination [36,37,38,39,40,41,42,43,44,45,46,47,48].

Surgery for resectable pancreatic cancer is associated with high perioperative morbidity. Oncological surgeons might think that adding HIPEC would result in higher morbidity, longer hospital stays, and higher hospital costs. However, as we can see from our results, there is no significant difference in length of hospital stay, Clavien–Dindo complications, or cost between patients who have HIPEC and those who do not.

We hypothesize that PaCSCs, after epithelial-mesenchymal transition, may migrate into blood vessels, potentially contributing to occult systemic micrometastasis and causing distant disease recurrence. Interestingly, we saw no effect on distal recurrence between the two groups, but locoregional recurrence was significantly lower in HIPEC patients than in control patients, *p*-value of 0.022. Possibly, administration of adjuvant HIPEC treatment would help control locoregional disease

We are the first group to investigate, in a clinical trial, how HIPEC controls pancreatic cancer and its PaCSCs. Although many markers of PaCSCs have been identified, a universal marker to identify them is still lacking, so the combination of several markers could increase the purity of isolated PaCSCs [49,50]. Since the surface markers CXCR4 and CD133 are not only expressed on pancreatic tumor stem cells but also on cells of the immune system and hematopoietic stem cells, respectively, molecular characterization of PaCSCs by flow cytometry techniques using CXCR4^+^ and CD133^+^ double labeling from the selection of the epithelial tumor cell population (EpCAM^+^) was proposed; that is by triple EpCAM+CXCR4+CD133+ labeling, eliminating a non-specific background of other false positive cell types [51,52].

We initiated this project because we hypothesized that HIPEC with gemcitabine could reduce pancreatic cancer tumor progression by reducing the subpopulation of PaCSCs and improving overall survival. We determined the presence of PaCSCs from pre-HIPEC and post-HIPEC intraperitoneal residual solution samples. The findings of this study can validate our hypothesis about the effects of HIPEC against PaCSCs (*p*-value: 0.018) as well as the association between residual PaCSCs and poor prognosis in these patients, by detecting its effect on them at the level of locoregional recurrence, despite not find an effect on overall or disease-free survival, probably due to the short survival of this type of patients. This leads to increased consideration of undiagnosed intra-abdominal pancreatic tumor cells with respect to PaCSC isolation. Perhaps we need to add new methods to diagnose the intraoperative stage of the disease. Although these results are very promising, more studies will be necessary to know the exact prognostic value of these CSCs and the possibility after their isolation of identifying a selective chemotherapy for each patient.

Despite our results, the limitations of our study were the small population included, which we recruited only in our hospital. This was an open phase II-III study, but no other pancreatic or oncologic surgical units were added to our project. Possible causes for this were perhaps excessive rigidity in the boundaries of the field of activity between surgical units and the belief of excessive morbidity and cost of therapy. In addition, unfortunately, the SARS-CoV-2 pandemic paused recruitment and activity for months.

## 5. Conclusions

In conclusion, we obtained promising results after using HIPEC with gemcitabine as adjuvant treatment after surgery for pancreatic cancer. Locoregional recurrence was significantly lower when we used HIPEC with gemcitabine; moreover, PaCSCs associated with tumor recurrence decreased. Therefore, locoregional treatment in patients with pancreatic cancer could represent a promising strategy to eliminate PaCSCs that evade therapy, which are subsequently responsible for tumor recurrence and a poorer prognosis in patients with this disease. In addition, it was a safe and cost-effective technique.

## Figures and Tables

**Figure 1 cancers-16-01718-f001:**
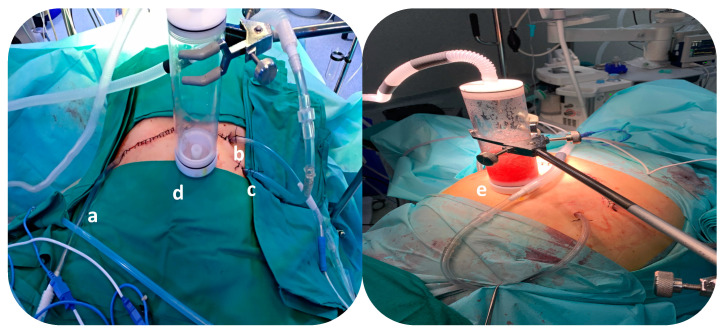
Closed hyperthermic intraperitoneal chemotherapy (HIPEC) system using CO_2._ a: Inflow catheter to fill the abdomen with the hyperthermic solution; b: Outflow catheter to remove the fluids into the HIPEC machine; c: Inflow catheter to create the turbulence with CO_2_; d: Gas Exchange System; e: During the recirculation of the drug, the gas exchanger allows the CO_2_ to be extracted, controlling the intra-abdominal pressure.

**Figure 2 cancers-16-01718-f002:**
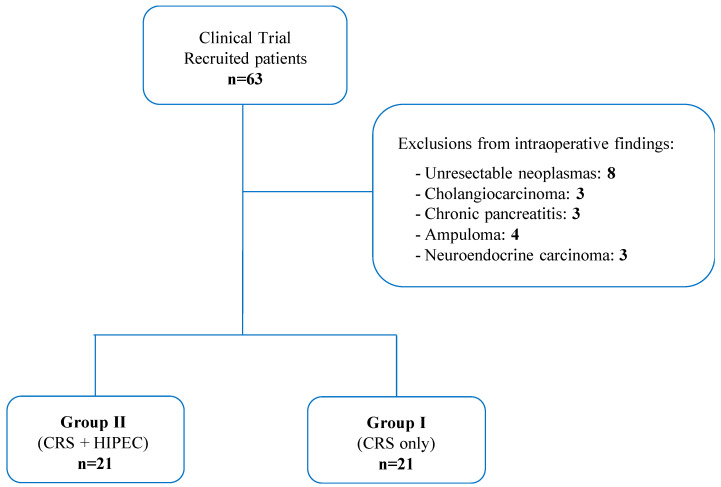
Recruitment of patients for the clinical trial according to inclusion and exclusion criteria. CRS: Cytoreductive surgery; HIPEC: Hyperthermic Intraperitoneal Chemotherapy.

**Figure 3 cancers-16-01718-f003:**
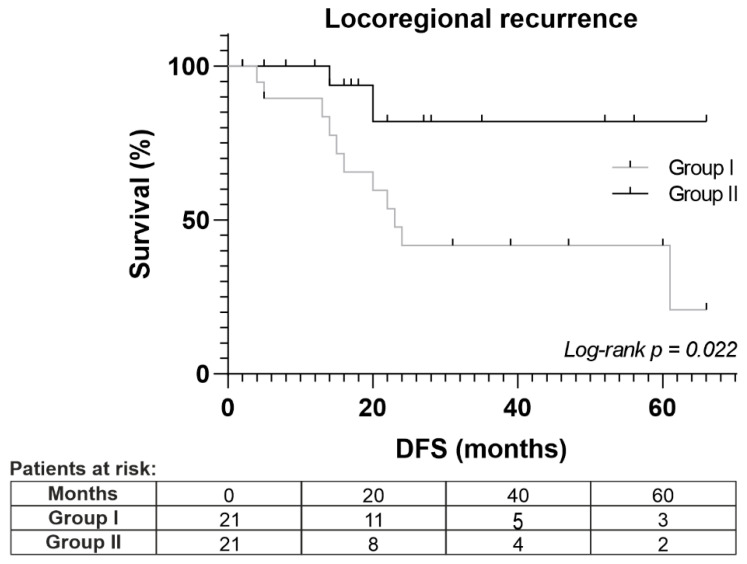
Locoregional recurrence analysis by Kaplan–Meier analysis between Groups I (control) and II (experimental group). DFS: Disease-free survival.

**Table 1 cancers-16-01718-t001:** Clinical, surgical, treatment, and histological characteristics of the patients. PaCSCs: Pancreatic cancer stem cells; HTA: Arterial hypertension; DM: Diabetes mellitus; DL: dyslipidaemia; TP: Total pancreatectomy; CDP: Cephalic duodenopancreatectomy; STP: Subtotal pancreatectomy; DP: Distal pancreatectomy; CCP: Corporocaudal pancreatectomy; Gem: Gemcitabine; FOL: FOLFIRINOX; Nab-P: Nab-paclitaxel; CAP: Capecitabine; TNM: Tumor node metastasis. *p*-value < 0.05, Group I (control) versus Group II (experimental group).

Characteristics	Group II (*n* = 21)	Group I (*n* = 21)	*p*-Value
Age (years, *M ± SEM*)	64 ± 1.7	69.3 ± 2	0.024 *
Overall survival (months, *median*)	17.1 (0.6–61.9)	18 (0.8–61.8)	0.899
Disease-free survival (months, *median*)	14 (2–66)	10 (2–66)	0.888
Operative time (minutes, *median*)	350 (280–420)	300 (120–360)	0.002 *
Hospital stay (days, *median*)	11 (7–28)	17 (4–69)	0.302
PaCSCs (CXCR4 + CD133 + EpCAM)	19.7 ± 8.2	13.8 ± 7.1	0.919
Sex	*n* (%)	*n* (%)	
Male	6 (29)	13 (62)	0.03 *
Female	15 (71)	8 (38)	
Symptoms			
HTA	10 (48)	9 (43)	0.757
DM	9 (43)	7 (33)	0.525
DL	10 (48)	6 (29)	0.204
Jaundice	17 (81)	12 (57)	0.095
Abdomnal pain	8 (38)	7 (33)	0.747
Constitutional syndrome	5 (24)	5 (24)	0.999
Resections			
TP without splenectomy	1 (5)	4 (19)	0.502
TP with splenectomy	9 (43)	5 (24)	
CDP	9 (43)	8 (38)	
STP	1 (5)	1 (5)	
DP	0	1 (5)	
CCP	1 (5)	2 (10)	
Adjuvant treatment			
No	9 (43)	5 (24)	0.190
Yes	12 (57)	16 (76)	
Chemotherapy regimen			
No	9 (43)	5 (24)	0.215
Gem	3 (14)	8 (38)	
FOL	2 (10)	4 (19)	
Gem + Nab-P	4 (19)	2 (10)	
CAP	2 (10)	0 (0)	
Gem + CAP	1 (5)	2 (10)	
Pancreatitis complications			
Delayed gastric emptying	2 (10)	2 (10)	0.999
Pancreatic fistula (B,C)	0	1 (5)	0.757
Hemorrhage	0	2 (10)	0.525
Clavien-Dindo			
I	4 (19)	3 (14)	0.804
II	5 (24)	3 (14)	
III	2 (10)	3 (14)	
IV	0	1 (5)	
V	1 (5)	1 (5)	
Differentiation			
Good	4 (19)	3 (14)	0.08
Moderate	7 (33)	14 (67)	
Poor	10 (48)	4 (19)	
Invasion			
Neurologic invasion	15 (71)	15 (71)	0.999
Vascular invasion	7 (33)	8 (38)	0.747
Lymphatic Invasion	7 (33)	11 (52)	0.212
TNM			
Ia	2 (10)	5 (24)	0.419
Ib	6 (29)	3 (14)	
IIa	1 (5)	2 (10)	
IIb	6 (29)	8 (38)	
III	6 (29)	3 (14)	
Locoregional recurrence	2 (10)	11 (52)	0.004 *
Distant recurrence			
No	12 (57)	14 (62)	0.757
Liver	7 (33)	8 (38)	
Lung	2 (10)	0 (0)	
Mortality (>30 days)	13 (62)	13 (62)	0.999

* *p* < 0.05, tumor versus non-tumor tissue.

## Data Availability

The data presented in this study are available in this article.

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
