# Peer review of "Safety and Effectiveness of Perioperative Hyperthermic Intraperitoneal Chemotherapy with Gemcitabine in Patients with Resected Pancreatic Ductal Adenocarcinoma: Clinical Trial EudraCT 2016-004298-41"

_cancers, 2024, doi:10.3390/cancers16091718_

Round 1
Reviewer 1 Report
Comments and Suggestions for Authors
The authors conducted the prospective randomized trial to evaluate the efficacy of HIPEC during the resection of pancreatic cancer on long-term outcomes. They included 42 patients who underwent resection of pancreatic cancer and the half received HIPEC and the other half completed the standard radical resection. They demonstrated a statistically significant reduction of locoregional recurrence among the patients who received HIPEC compared to those in the control group. They also investigated the amount of pancreatic cancer stem cells (PaCSCs) before and after HIPEC and they speculated the association between the reduction of PaCSCs by HIPEC and improved locoregional-recurrence free survival.
Although the topic and idea of the treatment was very interesting, there were too many scientific flaws in the study design, and it was difficult to interpret the data appropriately. Thus, their conclusion appeared confusing. Here were my comments and concerns:
1. Power calculation: the primary outcomes and their assumption to determine the sample size should have been clearly described in the method section. They listed multiple outcomes as "main outcomes" and it was not clear whether no difference in OS/DFS among the groups indicated true absence in efficacy of HIPEC or "under-power" of the study.
2. While they described the patients were assigned after R0 resection to either HIPEC or non-HIPEC group (page 3). I think the surgical margin could not be fully accessed during the resection and this description was inaccurate. Furthermore, final surgical margin status was not shown in Table 1, which made the interpretation of the study result impossible in terms of the impact of HIPEC on the locoregional recurrence.
3. No data for post-operative therapy (adjuvant chemo- or chemoXRT) were not shown.
4. For the amount of PaCSCs, there were no data shown in the control group (value in the end of operation), and thus, it was not clear if HIPEC was the primary cause of the reduction of PaCSC after HIPEC.
5. HIPEC is in general considered as the therapy aiming to control the peritoneal metastatic cancer cells, and their conclusion did not make sense that HIPEC can reduce locoregional recurrence, but not distant recurrence. They should have shown the detail of the systemic recurrence. What was the impact of HIPEC on peritoneal recurrence?
Reviewer 2 Report
Comments and Suggestions for Authors
I am grateful for the opportunity to review the paper “Safety and effectiveness of perioperative hyperthermic intraperitoneal chemotherapy with gemcitabine in patients with resected pancreatic ductal adenocarcinoma. Clinical trial EudraCT 2016-004298-41". Your study investigated randomized control trial of HIPEC + GEM for PDAC patients. This research gives us the future perspective, and this addresses a significant clinical concern. In summary, this results seem worth noting and the evaluation methods using pancreatic cancer stem cell is novel. However, I do have a few suggestions and concerns.
Major points
# Throughout the manuscript, many errors and typos were found, making understanding the results difficult. For example, “Exclusion criteria” and “Data collection” were just a list of items, not a complete sentence. Please look over it and make corrections accordingly.
# Some of the important variables didn’t include this analysis. Is there any discrepancy in the level of CA19-9? Also, the authors should mention NCCN resectability for all patients, as well as the location of tumor and lymph node metastases.
# For me, the meaning of Table 2 seems obscure. What is the significance of this? I assume that Table 2 is too long. So I recommend this should be depicted as supplementary file.
Minor points
# All 42 patients had R0 resection? What is the definition of R0 resection – 0mm rule or 1mm rule?
# There were two people experiencing in-hospital mortality. What were the cause of death?
# The authors should describe the actual number of significance, not summarizing as “ns”.
Comments on the Quality of English LanguageModerate revision will be needed.
Reviewer 3 Report
Comments and Suggestions for Authors
The paper is an interestic phase II-III RCT study reporting results of CRS with or without HIPEC with Gemcytabine for pancreatic adc.
However, many limitations are present and prevent to draw a conclusion: the sample is very small, and the mean FU is only 18 months...
1. The authors report no difference in distant recurrence, but a significant lower local recurrence in HIPEC group. So, how can the OS be the same in the 2 groups?
2. How many patients have both local and distant recurrence, if any?
3. Which are the sites of distant recurrence, and does it affects the survival?
4. Is the trial still ongoing or is closed? Perhaps it would be better to wait for a longer mean FU and to recruit more patients.
5. I am not sure that the conclusions can be supported by the results reported.
Reviewer 4 Report
Comments and Suggestions for Authors
It is a good, documented and original article in the same time that I think deserves to be published.
I would like the Conclusions chapter to be individualized as introduction, material and method, results and discussions.
Round 2
Reviewer 1 Report
Comments and Suggestions for Authors
Thank you for your responses to my comments. The revised manuscript remained confusing to me and several fundamental things related to study design needs to be clarified.
1. They did not describe the power calculation for the entire study. They stated that the sample size was set as "to detect a statistically significant difference between the two proportions". What did they intend to detect as the primary outcomes? On the other hand, they mentioned no difference in OS/DFS was likely due to underpower. If they set the sample size to detect the difference in OS, this comment was inadequate.
2. Was HIPEC given in the separate index operation from the definitive resection? The description for the procedure of HIPEC was very confusing and it was hard to understand the sequence of treatment the patients received. Given the 1mm R0 rule was applied to the inclusion criteria of the study, I assume all patients completed resection surgery, waited for the final pathology that proved negative margins (this diagnosis cannot be made intraoperatively), and then returned to the OR for HIPEC. How long the interval from the initial resection and HIPEC procedure? I don't think HIPEC at weeks after resection is adequate because the efficacy to perfuse the chemotherapy in the peritoneal space will be significantly compromised due to post-operative adhesion. If the patients received HIPEC simultaneously during the initial resection (this sounds a more reasonable practice to me), the description of the inclusion criteria must be incorrect.
Author Response
Reviewer 1
- They did not describe the power calculation for the entire study. They stated that the sample size was set as "to detect a statistically significant difference between the two proportions". What did they intend to detect as the primary outcomes? On the other hand, they mentioned no difference in OS/DFS was likely due to underpower. If they set the sample size to detect the difference in OS, this comment was inadequate.
For the calculation of the sample size estimate, a prevalence of 2.7% was considered for pancreatic cancer and a population of approximately 260,000 inhabitants in the population of Ciudad Real Integrated Care Management. A confidence level of 95% was assumed, with a 5% alpha risk and a 20% beta risk. With these parameters, a sample size of at least 41 individuals was obtained so that the results could be extrapolated to the population. Therefore, 21 patients per group were included. This study was supervised by statisticians from the methodological support service of our hospital.
The main variables that we wished to consider with prognostic value were overall survival, disease-free survival, locoregional recurrence, distant recurrence, PaCSCs levels. Regarding the comment we made about the absence of significant differences between groups in terms of OS and DFS, we think that this is due to the fact that pancreatic cancer is a disease with a low survival time, which makes it difficult to find significant differences in groups. Sorry for the inadequate comment.
- Was HIPEC given in the separate index operation from the definitive resection? The description for the procedure of HIPEC was very confusing and it was hard to understand the sequence of treatment the patients received. Given the 1mm R0 rule was applied to the inclusion criteria of the study, I assume all patients completed resection surgery, waited for the final pathology that proved negative margins (this diagnosis cannot be made intraoperatively), and then returned to the OR for HIPEC. How long the interval from the initial resection and HIPEC procedure? I don't think HIPEC at weeks after resection is adequate because the efficacy to perfuse the chemotherapy in the peritoneal space will be significantly compromised due to post-operative adhesion. If the patients received HIPEC simultaneously during the initial resection (this sounds a more reasonable practice to me), the description of the inclusion criteria must be incorrect.
Indeed, an R0 resection is an inclusion criterion, so all patients underwent this type of surgery. Once the tumor was removed, during surgery, an intraoperative diagnostic study was carried out by the pathologists of the Anatomical Pathology Service. If in this first study they confirmed the diagnosis of adenocarcinoma and that the surgical margins were free of disease in principle, in this case, the result was immediately communicated to the surgery service and the patient proceeded to receive gemcitabine by means of the HIPEC technique. All this was done during the course of the surgical process, without the patient leaving the operating room. The time that goes between R0 resection and gemcitabine recirculation ranges between 30-45 minutes. If, on the other hand, the pathologists determined that the resection performed was not R0 and that there were tumor-positive margins, a more aggressive surgery was performed again.
In fact, the patients received HIPEC simultaneously during the initial resection (to clarify the latter we have added a sentence in material and methods). After some time, when the pathologists made the final diagnosis, we confirmed the intraoperative diagnosis. If R0 had not been confirmed at this second check, this patient would not have been considered for the study. The patient knew and understood this situation because it was explained to him previously.
Reviewer 2 Report
Comments and Suggestions for Authors
Thank you for working on the comments. I believe that this paper meets the publication quality.
Comments on the Quality of English LanguageSome minor editing will be helpful. I think some of the words seem awkward.
Author Response
Reviewer 2
Thank you for working on the comments. I believe that this paper meets the publication quality. Some minor editing will be helpful. I think some of the words seem awkward.
Thank you very much for your comments. We have reviewed the wording and made changes to some words and phrases, which we have highlighted in green. We apologize if the English translation in this regard was initially not correct.
Reviewer 3 Report
Comments and Suggestions for Authors
The authors addressed many of the critical points after the original submission. Just a few observations:
1. In Fig. 1 the 2 groups are called Control Group and Experimental Group. Just after, in Table I, the 2 groups are called Group I and Group II. The authors could better explain the difference in the 2 groups, the reader has to go read the materials and methods section to find out which of the groups is the control and the experimental.
2. In Table I I would separate the clinical characteristics from the results (Clavien-Dindo complications, pancreatitis, local and distance recurrence, mortality). Perhaps the results should be on a separate table.
